# Social networks predict the life and death of honey bees

Benjamin Wild [1,8✉], David M. Dormagen [1,8], Adrian Zachariae[2,8], Michael L. Smith [3,4,5], Kirsten S. Traynor [1,6], Dirk Brockmann[2,7], Iain D. Couzin [3,4,5] & Tim Landgraf [1✉]

In complex societies, individuals' roles are reflected by interactions with other conspecifics. Honey bees (*Apis mellifera*) generally change tasks as they age, but developmental trajectories of individuals can vary drastically due to physiological and environmental factors. We introduce a succinct descriptor of an individual's social network that can be obtained without interfering with the colony. This 'network age' accurately predicts task allocation, survival, activity patterns, and future behavior. We analyze developmental trajectories of multiple cohorts of individuals in a natural setting and identify distinct developmental pathways and critical life changes. Our findings suggest a high stability in task allocation on an individual level. We show that our method is versatile and can extract different properties from social networks, opening up a broad range of future studies. Our approach highlights the relationship of social interactions and individual traits, and provides a scalable technique for understanding how complex social systems function.

[1] Department of Mathematics and Computer Science, Freie Universität Berlin, Berlin, Germany. [2] Robert Koch Institute, Berlin, Germany. [3] Department of Collective Behaviour, Max Planck Institute of Animal Behavior, Konstanz, Germany. [4] Centre for the Advanced Study of Collective Behaviour, University of Konstanz, Konstanz, Germany. [5] Department of Biology, University of Konstanz, Konstanz, Germany. [6] Global Biosocial Complexity Initiative, Arizona State University, Tempe, FL, USA. [7] Institute for Theoretical Biology, Humboldt University Berlin, Berlin, Germany. [8]These authors contributed equally: Benjamin Wild, David M. Dormagen, Adrian Zachariae. ✉email: b.w@fu-berlin.de; tim.landgraf@fu-berlin.de

In complex systems, intricate global behaviors emerge from the dynamics of interacting parts. Within animal groups, studying interactions helps to elucidate the individuals' functions[1–4]. Descriptors of individuals derived from social interaction networks have been used to investigate, for example, pair bonding[5], intergroup brokering[6], offspring survival[7], cultural spread[8,9], policing behavior[10], leadership[11–13], organization of food retrieval[14], the ability to affect behavioral change[15], and behavior during famine events[16]. As our ability to collect detailed social network data increases, so too does our need to develop tools for understanding the significance and functional consequences of these networks[17].

Social insects are an ideal model system to study the relationship between social interactions and individual roles because task allocation has long been hypothesized to arise from interactions[18–20]. The relationship of individual roles within the colony and the social network; however, is not well understood. Individuals, for example, can modify their behavior based on nestmate interaction[21–24], and interactions change depending on where and with whom individuals interact[14,22,25,26]. These studies typically target specific types of interactions (e.g., food exchange), specific roles within task allocation (e.g., foraging), or specific stimuli within the nest (e.g., brood), but an automatic observation system could capture behaviors and interactions within a colony more comprehensively and without human bias. Measuring the multitude of social interactions and their effect on behavior, and the social networks over the lifetime of individuals without interfering with the system (e.g., by removing individuals) is an open problem.

In honey bees, task allocation is characterized by temporal polyethism[27–29], where workers gradually change tasks as they age: young bees care for brood in the center of the nest, while old bees forage outside[30,31]. Previous works often used few same-aged cohorts resulting in an unnatural age distribution[27,28,30,32]. The developmental trajectory of individuals can, however, vary drastically due to internal factors (i.e., genetics, ovary size, sucrose responsiveness[33–39]), nest state (i.e., amount of brood, brood age, food stores[40–42]), and the external environment (i.e., season, resource availability, forage success[43–46]). These myriad influences on maturation rate are difficult to disentangle, but all drive the individual's behavior and task allocation. Due to the spatial organization of honey bee colonies, task changes also result in a change of location, with further implications on the cues that workers encounter[31]. How and when bees change their allocated tasks in a natural setting has typically been assessed through destructive sampling (e.g., for measuring hormone titers of selected individuals), but understanding how all these factors combine would ideally be done in an undisturbed system.

With the advent of automated tracking, there has been renewed interest in how interactions change within colonies[47,48], how spatial position predicts task allocation[49], and how spreading dynamics occur in social networks[32]. Despite extensive work on the social physiology of honey bee colonies[50], few works have studied interaction networks from a colony-wide or temporal perspective[32,51]. While there is considerable variance in task allocation, even among bees of the same age, it is unknown to what extent this variation is reflected in the social networks. In large social groups, like honey bee colonies, typically only a subset of individuals are tracked, or tracking is limited to short time intervals[28,47,52,53].

Tracking an entire colony over a long time would allow one to investigate the stability of task allocation. Prior research has shown that during each life stage, an individual spends most of its time in a specific nest region[31,54], interacting with nestmates, but with whom they interact may depend on more than location alone (e.g., previous interactions, or the genetic diversity within the colony[55,56]). Social interactions permit an exchange of information and can have long-term effects on an individual's behavior[57]. While honey bees are well known for their elaborate social signals (e.g., waggle dance, shaking signal, stop signal[58–60]), they also exchange information through food exchange, antennation, or simple colocalization[61,62]. However, identifying an individual bee's role in a colony based on its characteristic patterns of interaction remains challenging, particularly with large numbers of individuals and multiple modes of interaction.

In this work, we investigate the relationship between an individual's social network and its lifetime role within a complex society. We developed a tracking method for unbiased long-term assessment of a multitude of interaction types among thousands of individuals of an entire honey bee colony with a natural age distribution. We introduce a low-dimensional descriptor, network age, that allows us to compress the social network of all individuals in the colony into a single number per bee per day. Network age, and therefore the social network of a bee, captures the individual's behavior and social role in the colony and allows us to predict task allocation, mortality, and behavioral patterns such as velocity and circadian rhythms. Following the developmental trajectories of individual honey bees and cohorts that emerged on the same day reveals clusters of different developmental paths, and critical transition points. In contrast to these distinct clusters of long-term trajectories, we find that transitions in task allocation are fluid on an individual level. We show that the task allocation of individuals in a natural setting is stable over long periods, allowing us to predict a worker's task better than biological age up to 1 week into the future.

## Results

**What is network age?** To obtain the social network structure over the lifetime of thousands of bees, we require methods that will track the tasks and social interactions of many individuals over consecutive days. We video recorded a full colony of individually marked honey bees (*Apis mellifera*) at 3 Hz for 25 days (from 1 August 2016 to 25 August 2016) and obtained continuous trajectories for all individuals in the hive[63,64]. We used a two-sided single-frame observation hive with a tagged queen and started introducing individually tagged bees into the colony ~1 month before the beginning of the focal period (see "Methods: 'Recording setup, data extraction, and preprocessing'" for details). To ensure that no unmarked individuals emerged inside the hive, we replaced the nest substrate regularly (approx. every 21 days). In total, we recorded 1920 individuals aged from 0 days to 8 weeks.

A worker's task and the proportion of time she spends in specific nest areas are tightly coupled in honey bees[31]. We annotated nest areas associated with specific tasks (e.g., brood area or honey storage) for each day separately (see "Methods: 'Nest area mapping and task descriptor'"), as they can vary in size and location over time[65]. We then use the proportion of time an individual spends in these areas throughout a day as an estimate of her current tasks.

We calculated daily aggregated interaction networks from contact frequency, food exchange (trophallaxis), distance, and changes in movement speed after contacts (see "Methods: 'Social networks'"). These networks contain the pairwise interactions between individuals over time. For each day and interaction type, we extract a compact representation that groups bees together with similar interaction patterns, using spectral decomposition[66,67]. We then combine each bees' daily representations of all interaction types and map them to a scalar value (network age) that best reflects the fraction of time spent in the task-associated areas using CCA (canonical-correlation analysis; refs. [68,69]). Note that network age is solely a representation of the social network and not of location; the fraction of time spent in the task-associated areas is

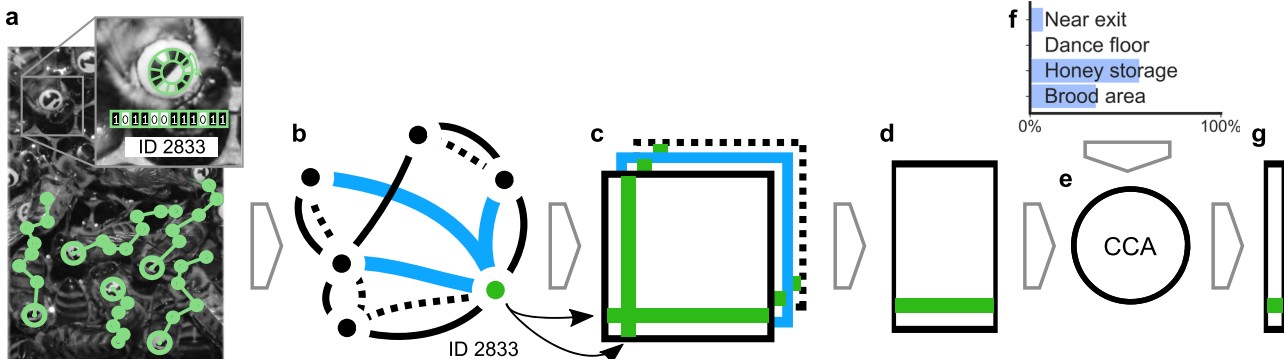

**Fig. 1 Network age, a one-dimensional descriptor of an individual's role within the colony, based on an individual's interaction pattern.** Using the BeesBook automated tracking system, we obtain lifetime tracking data for individuals (**a**). These tracks are used to construct multiple weighted social interaction networks (**b**). We aggregate daily networks (**c**) to then extract embeddings that group bees together with similar interaction patterns, using spectral decomposition (**d**). Finally, we use a linear transformation (**e**; CCA canonical-correlation analysis) that maximizes correlation with the fraction of time spent in different nest areas (**f**) to compress them into a single number per day called "network age" (**g**).

only used to select which information to extract from the social networks (e.g., by assigning higher importance to proximity contacts, see Supplementary Note 1.1). Network age can still represent an individual's location, but only if this information is inherently present in the social networks. Network age thus compresses millions of data points per individual and day (1919 potential interaction partners, each detected 127,501 ± 50,340 times on average per day, with four different interaction types) into a single number that represents each bee's daily position in the multimodal temporal interaction network. Since CCA is applied over the 25 days of the focal period, network age can only represent interaction patterns that are consistent over time. See Fig. 1 for an overview and "Methods: 'Network age: from networks to spectral embeddings to CCA'" for a detailed description of the methods.

Network age is a unitless descriptor. We scale it such that 90% of the values are between 0 and 40 to make it intuitively comparable to a typical lifespan of a worker bee during summer, and because biological age is commonly associated with task allocation in honey bees. This scaling can be omitted for systems where behavior is not coupled with biological age.

**Network age correctly identifies task allocation.** Because of the inherent coupling of tasks and locations in a honey bee colony, we expect a meaningful measure of social interaction patterns to be correlated with the individual's spatial preferences. We quantify to what extent network age captures this correlation by using multinomial regression to predict the fraction of time each bee spends in the annotated nest areas (see "Methods: 'Task prediction models and bootstrapping'"). Note that while we also used these spatial preferences to select which information to extract from the interaction networks, it is not certain whether the spatial information is contained in the social network in the first place, and how well a single dimension can capture it. Furthermore, the social network structure could vary over many days with changing environmental influences, preventing the extraction of a stable descriptor. The regression analysis allows us to compare different variants of network age to biological age as a reference.

To evaluate the regression fit, we use McFadden's pseudo $R^2$ scores $R^2_{McF}$[70]. Network age is twice as good as biological age at predicting the individuals' location preferences, and therefore their tasks (network age: median $R^2_{McF} = 0.682$, 95% confidence interval (CI) [0.678, 0.687]; biological age: median $R^2_{McF} = 0.342$, 95% CI [0.335, 0.349]; 95% CI of effect size [0.332, 0.348], $N = 128$; likelihood ratio $\chi^2$ test $p \ll 0.001$, $N = 26403$, Supplementary

Table 2 and see "Methods: 'Statistical comparison of models'" for details). Network age provides a better separability of time spent in task-associated nest areas than biological age (Fig. 2a, example cohort in Fig. 2c, Supplementary Note 1.2 for all cohorts). Network age correlates with the location because of the inherent coupling between tasks and nest areas. Still, it is not a direct measure of location: bees with the same network age can exhibit different spatial distributions and need not directly interact (see Supplementary Note 2.1).

While we can improve the predictive power of network age by extracting a multidimensional descriptor instead of a single value (see "Methods: 'Network age: from networks to spectral embeddings to CCA' and 'Task prediction models and bootstrapping'" for details), the improvements for additional dimensions are marginal compared to the difference in predictiveness between the first dimension of network age and biological age (see Supplementary Table 2). This implies that a one-dimensional descriptor captures most of the information from the social networks that are relevant to the individuals' location preferences and therefore their tasks.

We experimentally demonstrated that network age robustly captures an individual's task by setting up sucrose feeders and identifying workers that foraged at the feeders (known foragers, $N = 40$, methods in "Methods: 'Forager groups' experiment'"). We then compared the biological ages of these known nectar foragers to their network ages. We made these two quantities comparable by z-transforming them because they do not have the same unit of measure. As expected, foragers exhibited a high biological age and a high network age, whereas biological age exhibited significantly larger variance than network age (Fig. 2b; Levene's test[71], performed on z-transformed values: $p \ll 0.001$, $N = 200$). Indeed, while we observed a forager as young as 12 days old, that individual had a network age of 25.5, demonstrating that network age more accurately reflected her task than her biological age (z-transformed values: biological age $-0.46$; network age 0.61).

Tagging an entire honey bee colony is laborious. However, by sampling subsets of bees, we find that network age is still a viable metric, even when only a small proportion of individuals are tagged and tracked. With only 1% of the bees tracked, network age is still a good predictor of task (median $R^2_{McF} = 0.516$, 95% CI [0.135, 0.705], $N = 128$) while increasing the number of tracked individuals to 5% of the colony results in an $R^2_{McF}$ value comparable to the fully tracked colony (5% of colony tracked: median $R^2_{McF} = 0.650$, 95% CI [0.578, 0.705], $N = 128$; whole

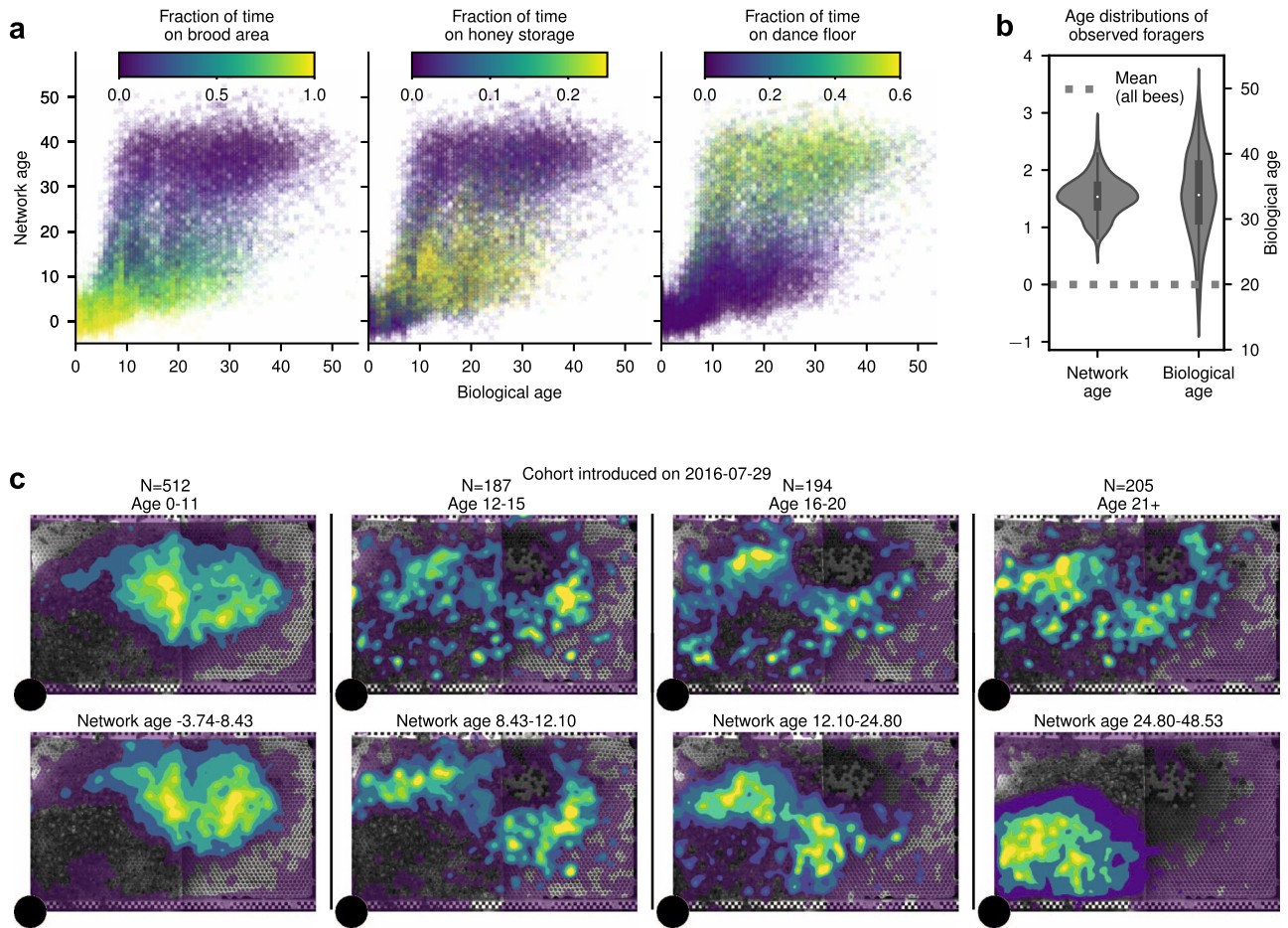

**Fig. 2 Network age is an accurate descriptor of task allocation. a** The proportion of time spent on task-associated locations in relation to biological age and network age with each cross representing one individual on one day of her life. For a given value on the *y*-axis (network age) colors are more consistent than for a given value on the *x*-axis (biological age). **b** *Z*-transformed age distributions for known foragers visiting a feeder (*N* = 40 observed individuals). The variance in biological age is greater than the variance in network age (boxes: center dot; median; box limits; upper and lower quartiles; whiskers, 1.5× interquartile range). Corresponding biological ages are also shown on the right *y*-axis (original biological age: 34.2 ± 7.9; original network age: 38.3 ± 4.6; mean ± standard deviation). **c** Spatial distributions of an example cohort over time (bees emerged on 29 July 2016, 64 individuals over 25 days), grouped by biological age (top row) versus network age (bottom row). Note how network age more clearly delineates groups of bees than biological age, with bees transitioning from the brood nest (center of the comb), to the surrounding area, to the dance floor (lower left area). The shaded areas depict density percentiles (brightest to darkest: 99%, 97.5%, 95%, 80%, 70%, 20%).

colony tracked: median $R^2_{\mathrm{McF}} = 0.682$, 95% CI [0.678, 0.687], $N = 128$; see Supplementary Note 2.2). Similarly, we find that an approximation of network age can be calculated without annotated nest areas: Network age can be extracted in an unsupervised manner using principal component analysis (PCA) on the spectral embeddings of the different interaction type matrices (median $R^2_{\mathrm{McF}} = 0.646$, 95% CI [0.641, 0.650], $N = 128$, see "Methods: 'Network age: unsupervised variant using PCA'").

**Developmental changes over the life of a bee**. Network age reveals differences in interaction patterns and task allocation among same-aged bees (Fig. 3a). After around 6 days of biological age, the network age distribution becomes bimodal (see "Methods: 'Quantifying when bees first split into distinct network age modes'"). Bees in the functionally old group (high network age) spend the majority of their time on the dance floor, whereas same-aged bees in the functionally young group (low network age) are found predominantly in the honey storage area (Fig. 3a). Transitions from high to low network age are a rare occurrence in our colony (see Supplementary Note 2.3).

We attribute the split on the population level to distinct patterns of individual development. Clustering the time series of network ages over the lives of bees identifies distinct developmental paths within same-aged cohorts. We set the number of clusters to three as this is the minimum number of clusters that separates an early and a late transition from low to high network age in all tested cohorts (see "Methods: 'Network age transition clustering'" for further details). In the cohort that emerged on 1 August 2016, the first developmental cluster (blue, Fig. 3b) rapidly transitions to a high network age (likely corresponding to foraging behavior) after only 11 days. The second cluster (orange) transitions at ~21 days of biological age, while bees in the third cluster (green) remain at a lower network age throughout the focal period. We see similar splits in developmental trajectories for all cohorts, although the timing of these transitions varies (see "Methods: 'Network age transition clustering'" for additional cohorts). Such divergence in task allocation has been previously shown in bees; factors that accelerate a precocious transition to foraging include hormone titers[72], genotype[35], and physiology, especially the number of ovarioles[73], and sucrose response threshold[74].

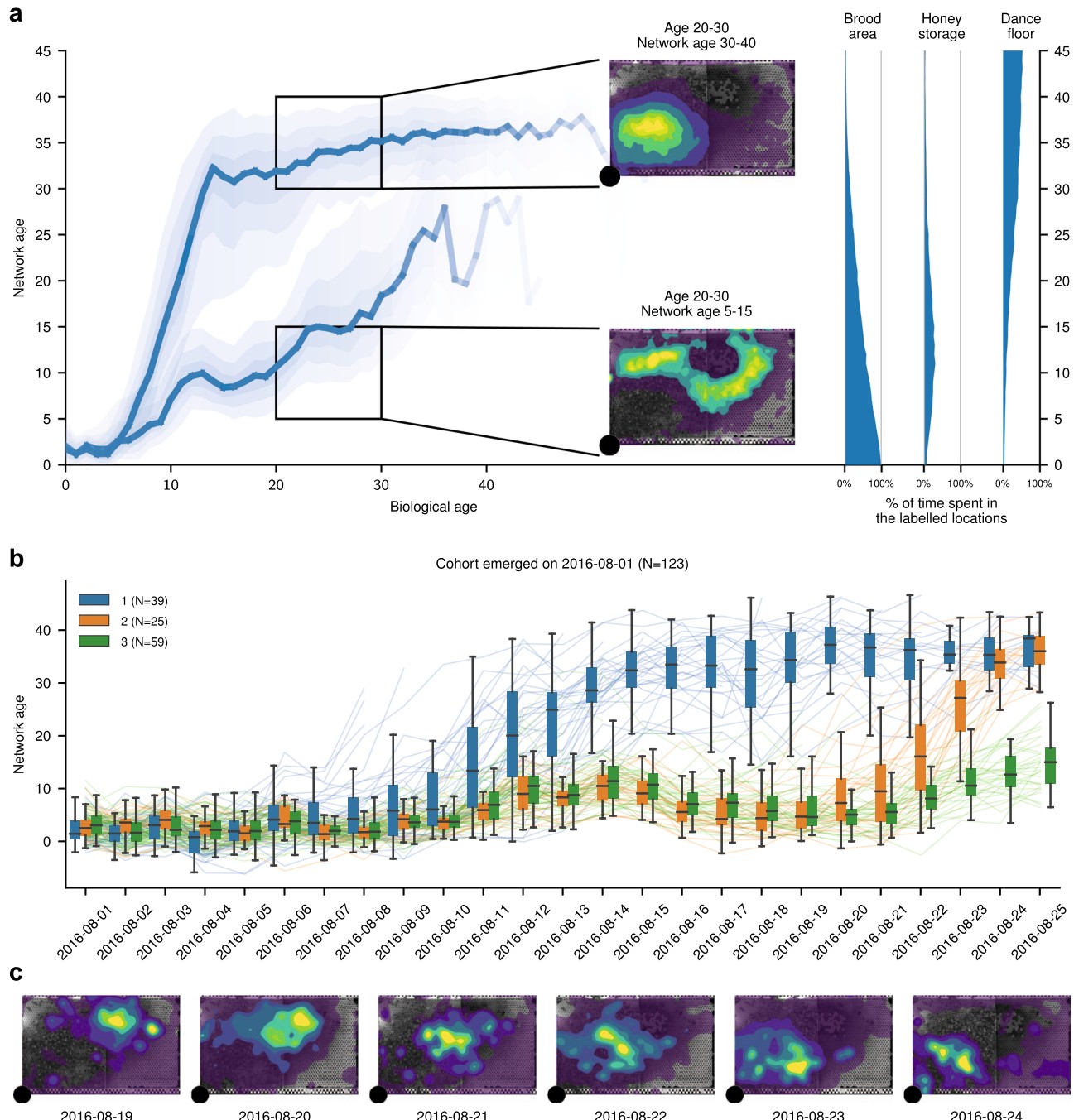

**Fig. 3 Network age reveals distinct developmental paths. a** Left: The median of network ages over biological ages for all individuals that lived more than 11 days split by a threshold on the age of 11 ($T = 23.07$, calculated using Otsu's method[84].The upper line contains all bees that fall above this threshold ($N = 832$), the lower contains all bees below that threshold ($N = 563$). The shaded areas depict 20%, 40%, 60% data intervals. We observe a split in network age corresponding to different tasks: The upper heatmap (network age 30–40, biological age 20–30, 577 bees, 857,283 data points) corresponds to the dance floor, while the lower heatmap (network age 5–15, biological age 20–30, 381 bees, 742,622 data points) borders between the dance floor and brood nest. Right: The mean fraction of time a bee with a given network age spends on our annotated regions throughout day. **b** Lines depict the network age of individual bees of a same-aged cohort with the colors indicating clusters of their network age over time. Boxes summarize bees belonging to each cluster for a given day (center line, median; box limits, upper and lower quartiles; whiskers, 1.5× interquartile range). **c** Heatmaps showing the spatial distribution of bees in the developmental cluster 2 (orange) from 19 August 2016 to 24 August 2016. The smooth transition in network age (orange in line plot, **b**) from one mode to another corresponds to a smooth transition in spatial location (heatmaps, **c**). The shaded areas depict density percentiles (brightest to darkest: 99%, 97.5%, 95%, 80%, 70%, 20%).

The transition from low to high network age over multiple days is characterized by a gradual shift in the spatial distribution (see example in Fig. 3c), highlighting that an individual's task changes gradually. The network age of most bees is highly repeatable

(median $R = 0.612$ 95% CI = [0.199, 0.982], see "Methods: 'Repeatability'" for details), indicating task stability over multiple days. Both findings (gradual change over a few days and high repeatability) are consistent with the dynamics of the underlying

physiological processes, such as vitellogenin and juvenile hormone, that influence task allocation and the transition to foraging[75].

**Network age predicts an individual's behavior and future role in the colony.** Network age predicts task allocation (i.e., in what part of the nest individuals will be) up to 10 days into the future. Knowing the network age of a bee today allows a better prediction of the task performed by that individual next week than her biological age informs about her current tasks (Fig. 4c, binomial test, $p \ll 0.001$, $N = 55390$, 95% CI of effect size [0.055, 0.090], $N = 128$, see "Methods: 'Future predictability'" for details). We confirm that this is only partially due to network age being repeatable (see "Methods: 'Future predictability'"). We do note, however, that our ability to predict the future tasks of a young bee is limited, especially before cohorts split into high and low network age groups (Fig. 3a). This limitation hints at a critical developmental transition point in their lives, an attractive area for future study.

We explicitly optimized network age to be a good predictor of task-associated locations. However, we find that network age predicts other behaviors better than biological age, including an individual's impending death (network age: median $R^2 = 0.165$, 95% CI [0.158, 0.172], versus biological age: median $R^2 = 0.064$, 95% CI [0.059, 0.068]; 95% CI of effect size [0.037, 0.039], $N = 128$, likelihood ratio $\chi^2$ test $p \ll 0.001$, $N = 26{,}403$). Biologically young but network-old bees have a significantly higher probability of dying within a week (80.6% $N = 139$) than do biologically old but network-young bees (42.1% $N = 390$; $\chi^2$test of independence $p \ll 0.001$ $N = 529$; see Supplementary Note 3.1 for details). This is likely because a biologically young bee with a high network age, that is, a bee that starts to forage earlier in life and faces more perils imposed by the outside world, is more likely to die than a bee of the same age with a low network age. This finding is consistent with previous work showing increased mortality with precocious foraging[76,77].

We measure movement patterns of individual bees such as daily and nightly average speed, the circadian rhythm, and the time of an individual's peak activity. While these properties are related to task allocation due to the diurnal nature of foraging, they are not direct measures of an individual's location. Network age also captures these movement patterns better than biological

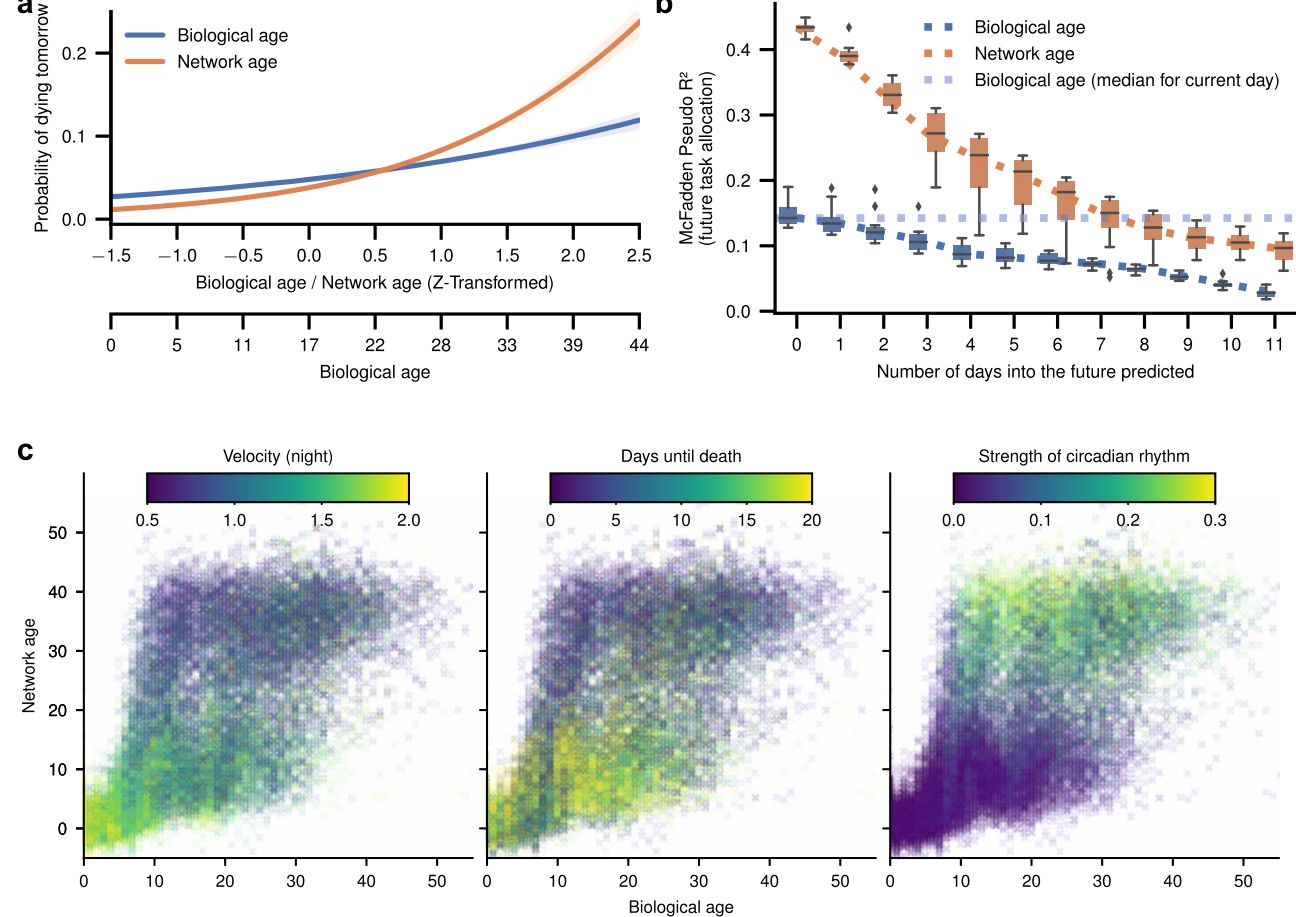

**Fig. 4 Network age can be used to predict other properties, such as mortality and circadian rhythms. It also predicts an individual's future task allocation. a** An individual's mortality on the next day based on her age (x-axis show original and z-transformed biological age and z-transformed network age). Bees with a low network age have lower mortality than biologically young bees; bees with a high network age have higher mortality than biologically old bees (shaded areas: 95% bootstrap confidence intervals for the regression estimates). **b** Network age can be used to predict task allocation and future behaviors. Network age predicts the task of an individual 7 days into the future better than biological age predicts the individual's task the same day (blue dotted line). Each box comprises $N = 12$ scores from models with $N = 12$ days of training data (center line, median; box limits, upper and lower quartiles; whiskers, 1.5× interquartile range; points, outliers). **c** Selected properties mapped for network age over biological age with each cross representing one individual on one day of her life. Note that for a given value on the y-axis (network age) colors are more consistent than for a given value on the x-axis (biological age).

age (likelihood ratio $\chi^2$ test $p \ll 0.001$, $N = 26403$, see Supplementary Table 1b for 95% CI of effect sizes).

To investigate whether network age is a good predictor of future task allocation and behavior only because it captures the spatial information contained in the social network, we repeat the analyses above using the time spent in task-associated locations as independent variables. We find that network age, even though it was extracted using this spatial information as a guide, is still a better predictor of an individuals' behavior (for all dependent variables likelihood ratio $\chi^2$ test $p \ll 0.001$, $N = 26403$, see Location (1D) in Supplementary Note 3.2 and Supplementary Table 1d for 95% CI of effect sizes). This difference in predictive power suggests that the multimodal interaction network contains more information about an individual than spatial information alone.

While we focus on predicting tasks from network age, we can control the information we extract from the observed social networks and derive variants of network age better suited for other research questions. By replacing the "task-associated location preferences" in the final step of our method with "days until death," we extracted a descriptor that captures social interaction patterns related to mortality. This descriptor improves the prediction of the individuals' death dates by 31% compared to network age (median increase in $R^2 = 0.05$; 95% CI [0.04, 0.06] $N = 128$, see "Methods: 'Targeted embedding using CCA'" and Supplementary Table 1c), opening up novel social network perspectives for studies such as the risk factors of disease transmission. Similarly, we extracted descriptors optimized to predict the movement patterns introduced in the last paragraph (for all except "Time of peak activity" likelihood ratio $\chi^2$ test $p \ll 0.001$, $N = 26,403$, see Supplementary Table 1c for 95% CI of effect sizes). These targeted embeddings provide precise control over the type of information we extract from the social networks and extend the network age method to address other important research questions in honey bees and other complex animal societies.

## Discussion

Combining automated tracking, social networks, and spatial mapping of the nest, we provide a low-dimensional representation of the multimodal interaction network of an entire honey bee colony. While many internal and external factors drive an individuals' behavior, network age represents an accurate way to measure the resulting behavior of all individuals in a colony noninvasively over extended periods.

We use annotated location labels to select which information to extract from the social network, but stress that network age can only contain information inherent in the social network. Therefore, the predictive power of network age demonstrates that the social interaction network by itself comprehensively captures an individual's behavior. We show that network age does not only separate bees into task groups, such as foragers and nurses, but also allows us to follow maturing individuals as they develop. A recent work derived a social maturity index in colonies of the social ant *Camponotus fellah*[78], highlighting a strong separation of nurses and foragers in the social network and high variability in transition timing. Similarly, network age is a fluid measure and the age at which individuals change between the task groups is highly variable. However, we find distinct clusters of developmental trajectories at the colony level, with some groups entering critical developmental transitions earlier in life than others. Further investigating the precise combination of internal and external factors that drive those transitions is a promising direction for future research.

These transition points are also reflected in changes in nest location, because spatial preferences, task allocation, and interactions are inherently coupled in honey bees. However, we show that network age is more than just a representation of location: Bees with the same network age do not necessarily share a location in the nest, and the time spent in task-associated locations is less predictive of an individual's current and future behavior than network age. In addition, we calculate a variant of network age that is not guided by auxiliary spatial information, but instead extracts the information with the highest variance from the social networks (Network age PCA). The PCA variant is still predictive of task allocation, suggesting that location is the predominant signal in the social network. However, the higher predictive power of the CCA network age variant and the targeted embeddings indicates that there is more information in the social network that our method can extract.

In this study, we extract network age from daily aggregated interaction networks, and thereby disregard potentially relevant intraday information. Furthermore, honey bees have a rich repertoire of interaction behaviors, of which we only capture a subset. The inclusion of intraday data or additional interaction types could reveal further differences between individuals (e.g., the temporal aspects of intraday interaction networks can disentangle the contribution of different modes of interactions[51]). While we study one colony in this work, we observe thousands of individuals and many overlapping cohorts. Our findings, in particular the existence of distinct developmental trajectories, and the fluidity and long-term stability in task allocation on an individual level, are consistent in all cohorts in our study. While some details, for example, the timing of developmental transitions, might depend on environmental circumstances, we believe that these results transfer to other colonies. There is no straight forward extension of the method to extract a common embedding of social networks that do not share individuals (e.g., over different experimental treatments or repetitions). Still, specific hypotheses can be tested using network age as long as a treatment group is compared to a control group within each trial. For example, while the meaning of specific values of network age can differ slightly between colonies, a group treated with pesticides could show differences in development, as measured by network age, relative to a control group from the same colony. Analyzing how network age changes within a day over other datasets with possibly other types of interactions, or how network age shifts in response to disease pressure or experimental manipulation of age demography would be potentially fruitful areas for future investigation, as previous work has shown that there is a relationship between pathogens and interaction behaviors[28,79–81].

Network age can be repurposed and extended for other research questions: We show that

(1) variants of network age capture different aspects from the social networks related to mortality, velocity, or circadian rhythms, and

(2) with a subsample of only 5% of the bees in the colony, we can extract a good representation of the social network.

This makes the method applicable to systems with far more individuals or with much less required experimental effort for a comparable number of individuals. Network age could be calculated in real time, opening up a wide range of possibilities for future research: For example, it would be possible to selectively remove bees that have just begun a developmental change to determine their influence on colony-wide task allocation. Sequencing individual bees could determine how known internal drivers of behavioral transition, like the double-repressor co-regulation of vitellogenin and juvenile hormone[33], are reflected in the social network. Our perspective captures both internal and external influences that impact social interactions and is thus applicable to all complex systems with observable

multimodal interaction networks. Network age can be adapted to questions that explore social interaction patterns independent of age and division of labor, making it broadly applicable to any social system. As such, our method will permit future research to analyze how complex social animal groups use and modify interaction patterns to adapt and react to biotic and abiotic pressures.

## Methods

**Recording setup, data extraction, and preprocessing.** We set up our observation hive on 24 June 2016, with a queen and ~2000 young bees (*A. mellifera*) sourced from a local host colony. To obtain newly emerged bees, we incubated brood from the host colony, and later from the observation colony in an incubator at 34 °C. Freshly emerged bees were marked every weekday. All bees were removed from the brood comb each day before marking, so the maximum age in each batch of bees was 24 h. After removing the hair from the bees' thoraxes with a wet toothpick, we applied shellac onto the thorax and attached a curved, circular tag. The number of bees marked per batch varied, but never exceeded 156. Marked bees were introduced to the colony through a backdoor entrance. After the initial marking period (27 days, starting 28 June 2016), the video recording was started on 24 July 2016 and stopped on 19 September 2016. Marking newborn bees continued approximately twice a week with the latest introduction on 23 August 2016. A total of 3166 individuals were marked. We recorded 1920 individuals from 30 cohorts during the focal period. See Supplementary Fig. 9 for the number of bees that were alive on each day. Bees had free access to the outside environment via a tube connected to the observation hive. We use the BeesBook[64] recording setup (scaffold, cameras, lighting, storage, marking procedure); however, for the experiments described here, we used custom-built IR flash circuit boards triggered by an Arduino controller, synchronized with the high-res cameras[82]. Combs were imaged at 3 Hz, alternating between sides of the observation hive, to avoid low contrast due to backlighting.

A total of 46 TB of video data were recorded and continuously moved to a network storage unit at the North-German Supercomputing Alliance (HLRN). After the recording season, the data were processed to detect and decode the bee markers[83] and track these detections through time[63]. The output data consists of timestamps, planar positions, three-dimensional rotations, IDs, and confidence scores for the decoded IDs.

Two cameras were used for each side of the nest with both viewports overlapping partially. Six reference points were marked on each comb side such that four points were visible in each of the cameras' recordings, with the center two points visible in both cameras. Reference points were identified and their image coordinates were extracted manually. The coordinates were then used to calculate the homography between the comb and the image plane. The homography was then used to rectify the tracking data that translated image coordinates to a metric reference frame, that is, the nest surface. See Supplementary Fig. 10 for a schematic of the setup.

Resulting tracking data were post-processed before entering the analysis. We discarded detections with low decoding confidence, that is, detections that the machine vision pipeline could not reliably decode. Remaining implausible detections (e.g., of IDs that had not been tagged yet) were removed in an additional filter step. The distribution of the number of detections of all IDs is strongly bimodal, the larger mode representing those bees that actually are in the observation hive, and the smaller mode representing erroneous decodings of tags that are not present on the given day. We use Otsu's method[84] to automatically determine the threshold which best separates those two modes and filter out all potentially incorrect IDs. The tracking data, therefore, contains gaps due to falsely filtering out correct detections, but also due to occlusions, such as when bees inspect cells or depart the nest on foraging trips.

**Forager groups' experiment.** Between 28 July 2016 and 22 August 2016, foragers were trained to a feeder (see Supplementary Fig. 11 for a photo of the feeder) offering unscented sucrose solution by gradually moving it from the colony (52.457032, 13.296635) to a sequence of locations "F1" to "F4" (see Supplementary Table 3). For days over which the feeder was moved, high sugar concentration was used and iteratively changed to control the number of new foragers. Once the final locations were reached, the feeder offered the highest concentration for 1–2 h per day. After a minimum of 3 days, training to the next location in the list was resumed. We photographed all bees landing at the location and manually transcribed the identity and time of arrival. A list of foragers visiting the feeder is given in Supplementary Table 4. The network age values around the day we first observed each bee at the feeding side is given in Supplementary Fig. 12.

**Bayesian lifetime model.** The death date of an individual could ideally be computed as the first date she was not detected in the hive. Unfortunately, this does not work in practice for two reasons. First, tags are sometimes incorrectly decoded, and because of the number of detections we have for each day, this means that most IDs will be detected at least a couple of times per day. Second, some bees were not visible at all on some days, even though they are not dead yet (see Supplementary Fig. 13 for an example).

We, therefore, use a Bayesian changepoint model to robustly estimate the death dates of all individuals. An individual is defined to be alive on all days since she emerged and was introduced into the colony (day $e$) up to the change point $d = e + l$, where $l$ is the number of days she was alive. We use a weakly informative prior $N(35, 50)$ for the number of alive days $l$. We model the probability that a bee is detected at least as often as a threshold $t$ while she is alive and less often than $t$ when she is not alive, using a Bernoulli distribution. We use a $Beta(5, 1)$ prior for this probability because we know that, typically, an alive bee will have many detections. For the threshold $t$ we use an informative $Beta(25, 1)$ prior because we know that a dead bee will have very few detections, if any. Note that we normalize the detection counts to $[0, 1]$ when fitting the model, that is, for each bee we divide the counts of daily detections by the maximum count of detections of that tag over the entire recording period. We sample this model using pymc3 and the NUTS sampler[85]. For each bee, we compute 2000 tuning samples and 1000 samples. The date of death is determined using the mean of those last 1000 Monte Carlo samples.

### Social networks

*Proximity interaction network.* Two bees were defined to be in proximity if their tags were <2 cm from each other (~1.4 body lengths) over at least 0.9 s (three frames with our recording frame rate). We construct affinity networks based on the counts of these proximity interactions without taking the duration of each contact into account to reduce the effect of bees resting next to each other.

*Euclidean proximity networks.* Euclidean proximities were determined for each pair of bees when both bees were visible. The daily average distance $d$ between two individuals was then transformed to two affinity matrices, the first derived by applying a Gaussian similarity function $d' = (-d^2 / 2\gamma^2)$ with $\gamma = \max(D)/4$ and the second by subtracting from the maximum distance ($d' = \max(D) - d$). $D$ is the matrix of all daily average distances on the same day.

*Trophallaxis networks.* We constructed an interaction network representing trophallaxis interactions (food exchange). To filter our data to detect trophallaxis events, we use a two-step approach. We first use a fast logistic regression with low precision to discard most of the non-trophallaxis encounters. We then use a slower convolutional neural network to further refine the results with higher recall.

To train the two classification models, we manually labeled bee interactions in our dataset by observing video sequences. To increase the fraction of positive events, we queried our data for bees that are close to each other, and approximately facing each other. Note that we did not distinguish the directionality or different types of trophallactic interactions.

This ground truth data contains 140 trophallaxis events out of the distinct 2651 events in total. For some events, we annotated a begin and end timestamp and could, therefore, use multiple frames for the training. In total, we had 25,835 training samples, each consisting of a pair of bee IDs, a timestamp, and a label (trophallaxis/not trophallaxis).

Because the prefiltering of the training data can introduce a sampling bias, we created another test set by labeling all possible interactions in 33 randomly sampled frames, containing a total of 15 trophallaxis events and 39,051 negative events (we use every pair of bees with a thorax distance of ≤3 cm as a possible candidate). This test set represents our data distribution without any bias.

In the classification step, we look at all pairs of bees with their thoraxes at a distance between 0.731 and 1.204 cm (i.e.. the 99th percentile of the positive events in the training data) together in a frame. For a pair of bees $(i, j)$ we have the locations of the thorax on the hive in millimeters $(xy_i, xy_j)$ and their orientations $(\alpha_i, \alpha_j)$. We calculate the approximate head position $h_i$ as $xy_i + d*[\cos(\alpha_i), \sin(\alpha_i)]$, where $d = 3.19$ mm. We calculate their relative orientation as $[\cos(\alpha_i), \sin(\alpha_i)] \cdot [\cos(\alpha_j), \sin(\alpha_j)]^T$. We then perform logistic regression on the euclidean distance of their thorax locations, the euclidean distance of their head locations, and their relative orientation.

The logistic regression was trained on the manually labeled samples, setting the threshold to get a recall of 85% at a precision of 21% (on a 20% validation test). This regression discards 62% of the true-negative samples (i.e., the specificity). For the remaining data points, those that were classified as possible trophallaxis by the first classifier, we extract trajectories of both bees for ~5 s (15 frames) around the possible trophallaxis events. We then use a convolutional neural network, again trained on the manually labeled data.

Evaluated on the test set, the two combined filters yield a recall of 60% at a precision of 47%, discarding 99.97% of negative samples (i.e., the specificity).

*Interaction effect networks.* For each proximity interaction with a duration no >60 s (to exclude bees resting next to each other) and with a minimum gap of at least 5 s since the last interaction of the same two individuals, we compute the difference in mean velocities within 30 s time windows before and after the interaction. This is done for both partners, and so we derive four networks based on the mean and cumulative changes, each split into negative and positive values. We use separate networks for the positive and negative values because this allows us to define affinity matrices that can only have positive edge weights.

*Temporal aggregation and post-processing of networks*. Time-aggregated networks were constructed by defining the weighted edge strength as the number of times two individuals were in proximity or engaged in trophallaxis. The networks were aggregated over 24 h without overlap. Edges in both networks are undirected. For subsequent analyses, all networks were represented as a square adjacency matrix with each element $i$, $j$ representing the affinity of bee $i$ with bee $j$ on this day, given by the interaction mode (e.g., for the network of trophallaxis counts, a high value represents many trophallaxis interactions between the two individuals). Each matrix is then preprocessed using a rank transform and normalized such that 0 represents the lowest affinity and 1 the highest affinity. Ties are resolved by assigning the same rank to identical affinities.

**Nest area mapping and task descriptor**. We manually outlined the capped brood area and visible honey storage cells for every day in background images from 30 July 2016 until 5 September 2016. To obtain the open brood area, we calculated the area of the comb that would become capped within 8 days. We extracted the background images by extracting the first frame from every video we recorded over a specific day (approximately one image every 5.6 min), and then applying a rolling median filter with window size 10 to these images. We then calculated the modal pixel value, for every pixel, over all the median images. For each side of the comb, we stitched together the background images from the two cameras on that side.

To get the approximate location of the dance floor, we used the detected waggle runs of our waggle dance detection system[86] that had high confidence (≥0.9) (see Supplementary Fig. 14). As nearly all waggle detections happened on one side of the comb, we exclusively labeled this area as the dance floor. We then fitted an ellipse to the detections using scikit-image[87], which we scaled manually to not intersect with the exit area. The dance floor area was consistent throughout the experiment, so in cases where we did not have waggle dance data for a given day, we interpolated the dance floor area over the adjacent days. Finally, we used a kaiser window applied over the consecutive days (window size = 5, *Beta* = 5) to smooth the annotations. We considered the region 7.5 cm around the exit tube as the nest region close to the exit.

To generate a task descriptor for every bee, we fetched one high confidence detection (>0.9) per bee for every minute of a day. We then counted how many of these detections per bee fell into the annotated regions. Then we normalized these counts per bee to 1 by dividing through the sum. This descriptor, therefore, contains the fraction of time each individual spends in each of the annotated regions. Data points outside the annotated areas are ignored for this descriptor but are used in all other parts of this work. For all evaluations, we consider the brood area region to be the sum of the annotated open and closed brood cell regions. See Supplementary Fig. 14 for an example of the annotations.

**Network age: from networks to spectral embeddings to CCA**. Network age is derived from the raw interaction matrices using spectral decomposition and CCA. For each day and interaction mode, the graph of interactions between all bees that were alive (see "Methods: 'Bayesian lifetime model'" for the definition of alive bees) on that day is retrieved as an adjacency matrix as described in "Methods: 'Social networks.'"

For each preprocessed affinity matrix, spectral embeddings[66] are calculated using the Python package scikit-network. We compute the first eight embedding dimensions (see Supplementary Note 2.2 for an evaluation of the performance of different numbers of embeddings).

For the non-symmetric interaction effect matrices, we use bispectral decomposition[88] to obtain one set of embeddings each for the rows and for the columns, to represent the two directions of an interaction.

For different days the eigenvectors of the embeddings and therefore the embedding values themselves can have an inverted sign. To correct this, we flip the sign of the values if the Spearman correlation between consecutive days is negative.

For every day, we now have a high-dimensional embedding per bee. We reduce the dimensionality further by applying CCA. We use CCA to find a linear transformation of the network embeddings to a three-dimensional vector that maximizes the correlation to a projection of the bees' task descriptors (as introduced in "Methods: 'Nest area mapping and task descriptor'"). We use the CCA implementation in scikit-learn[89]. We use the first dimension of this vector as 'network age' throughout this paper, but also evaluate multidimensional variants (Network age 2D, Network age 3D) in "Methods: 'Task prediction models and bootstrapping,' 'Statistical comparison of models,' and 'Prediction of other behavior-related measures.'"

For every dimension and day, we use robust scaling based on the 5th and 95th percentile of the network age distribution. A network age of zero corresponds to the 5th percentile and 40 corresponds to the 95th percentile. This stabilizes the distribution over time and also maps the values to a range comparable with the biological age of honey bees. We note that this scaling slightly improves the prediction of task allocation, but that the method also works without it. We enforce that the 5th percentile of network age corresponds to bees with a lower biological age than the 95th percentile such that biological age and network age have the same directionality.

**Network age: unsupervised variant using PCA**. We also calculate a variation of network age that does not require the annotated location descriptors. Instead of applying CCA to the concatenated spectral embeddings (see "Methods: 'Network age: from networks to spectral embeddings to CCA'"), we instead use PCA to reduce the dimensionality. This unsupervised network age still predicts task allocation better than biological age (see Supplementary Note 2.4 and "Methods: 'Task prediction models and bootstrapping'" for details).

**Task prediction models and bootstrapping**. To evaluate how well biological age and the different variants of network age represent an individual's task allocation, we use these measures as features to predict the proportion of time individuals spend in the brood area, dance floor, honey storage and near the exit (see "Methods: 'Nest area mapping and task descriptor'" for details on the nest area mapping). We evaluate the areas individually and in combined models. We evaluate different complexities of models (linear versus nonlinear) and different independent variables (e.g., network age and biological age).

To test different complexities of the relationships, we evaluate both a generalized linear model (GLM, the default model) and a small neural network consisting of two fully connected layers (listed as "nonlinear" in Supplementary Table 2) for each of the combinations of independent and dependent variables. The hidden layer of the neural network has a dimensionality of 8 and uses tangens hyperbolicus as its nonlinearity.

To evaluate the performance of the models for each area separately, we select a sigmoid as the link function of the GLM and the activation function of the neural network's last layer. We then optimize and calculate the likelihood of the data assuming a binomial distribution.

We also evaluate both models to simultaneously predict all four values of an individual's task allocation distribution. To this end, we choose a softmax function as the link function of the GLM and the neural network's final activation function. We then optimize and calculate the likelihood of the data assuming a multinomial distribution.

For all the combinations of independent and dependent variables, we repeat the described procedure for 128 bootstrap samples. For each model, we retrieve the final likelihood of the data $L_1$. We use PyTorch[90] and the L-BFGS optimizer to obtain maximum-likelihood estimates of the models. We also always fit a null model only consisting of the intercepts and retrieve its likelihood $L_0$. For each model and bootstrap iteration, we calculate McFadden's pseudo $R^2$[70] as $R^2_{\mathrm{McF}} = 1 - ((L_1)/(L_0))$. We then calculate the median and 95% CIs from these bootstrap samples. See Supplementary Table 2 for an overview of the results for all evaluated models. We test the significance of these results separately with the tests described in "Methods: 'Statistical comparison of models.'"

**Statistical comparison of models**. We use bootstrapped CIs of the effect strength to investigate whether a model based on one feature (e.g., network age) explains the dependent variables (e.g., task allocation distributions) significantly better than the same model based on a different feature (e.g., biological age). In addition, we use a likelihood ratio $\chi^2$ test to answer whether one feature (e.g., network age) provides additional information over biological age in a combined model.

*Bootstrapped CIs of the effect strength*. We draw 128 bootstrap samples of the combined daily bee data. For each sample, we calculate either the McFadden's pseudo $R^2$ in the case of the task allocation models (see "Methods: 'Task prediction models and bootstrapping' and 'Future predictability'") or the $R^2$ in the case of the other measures (see "Methods: 'Prediction of other behavior-related measures'" and Supplementary Note 3.1) for both a model based on biological age and the independent variable we want to compare with (e.g., network age). For each of these paired samples, we calculate the difference in scores of the two models. From these 128 differences, we calculate a two-sided 95% CI of the effect strength. If the null hypothesis (difference in scores is zero or less) is not contained in the confidence interval, we can reject the null hypothesis at an alpha level of 2.5%.

*Likelihood ratio test*. As the likelihood ratio test requires a nested model for testing, we compare a model based solely on biological age with a model based on a combination of biological age and the independent variable we want to compare with (e.g., network age).

We fit each model to the data and calculate the likelihoods of the data under the fitted models ($L_1$ for the combined model and $L_0$ for the model based on biological age). The likelihood ratio is given by $LR = -2\ln(L_0/L_1)$. If the null hypothesis that the models are equal were true, $LR$ would approximately follow a $\chi^2$ distribution with $k$ degrees of freedom (with $k = 4$ in the case of the task allocation model from "Methods: 'Task prediction models and bootstrapping'" and $k = 1$ in case of the general regression model for "Methods: 'Prediction of other behavior-related measures'" and Supplementary Note 3.1). We use the cumulative density function of the $\chi^2$ distribution to calculate the $p$ value.

**Repeatability**. We calculate the repeatability $R$ of the network age consisting of repeated measurements over several days of an individual $I$ as $R(I) = \mathrm{Var}_p/(\mathrm{Var}_i + \mathrm{Var}_p)$, with $\mathrm{Var}_i$ being the variance of the network age of an individual

measured over the available days and $\text{Var}_p$ being the variance of the mean network ages of a control group. The control group consists of all bees inside the same age span as $I$ on all days on which the network age values for $I$ were collected. A repeatability close to 1 means that the individual variance is low compared to the population variance. A repeatability close to 0 means that the individual variance outweighs the population variance.

**Network age transition clustering**. In order to cluster the transitions of different individuals in a cohort, we first collect the network ages for every individual in a feature vector where each entry corresponds to the individual's network age for one day. We then do a linear inter- and extrapolation for missing values (e.g., due to absence or the individual dying). For the cohort of bees, we calculate the euclidean distance between each individual's feature vector. Then we perform a hierarchical clustering using Ward's method[91] using the Python library scikit-learn[85] and extract the first three clusters. See Supplementary Fig. 15 for an example of the clustering. See Supplementary Fig. 16 for the network age development of different cohorts and Supplementary Fig. 17 for all bees.

We fixed the number of clusters to three for visualization purposes as that is the minimum number that showed a lagged transition from low to high network age for all cohorts. In hierarchical clustering, cutting off the dendrogram of the agglomerative clustering at a deeper level and thus increasing the number of clusters will further subdivide the existing clusters. Supplementary Fig. 18 gives an example with $N = 5$ clusters.

**Quantifying when bees first split into distinct network age modes**. We used $K$-means to cluster the network age distribution of every day into two distinct clusters corresponding to the two modes. Then we check for every bee that we observed at least once as a young bee below the age of 6 ($N = 1079$) at which age she first gets assigned to the higher cluster (mean = 12.33, 95% CI = [6, 25.7], median = 11, $N = 572$). We ignore bees that are never assigned to the upper cluster (e.g., because we do not observe them for a long enough timespan). See Supplementary Fig. 19 for the distribution of biological ages.

**Definition of circadian rhythmicity**. The motion velocity of a bee was determined by dividing the euclidean distance between two consecutive detections by the time passed (a multiple of $\frac{1}{3}$ seconds). Any duplicate IDs were discarded. The velocity was median filtered with a kernel size of 3 to remove outliers. See Supplementary Fig. 20 for an example of the velocity of a specific bee over multiple days.

Lomb–Scargle periodograms were computed for all individuals remaining in the dataset at any time in the period from 20 July 2016 to 18 September 2016. For each day and individual, the Lomb–Scargle periodogram was calculated on the motion velocities over an interval of 3 days, that is, including the preceding and following day. The circadian activity was confirmed as strong peaks at a period of 1 day. Lomb–Scargle periodograms were computed using the Astropy package[92]. See Supplementary Fig. 20 for an example of a bee's velocity and the resulting Lomb–Scargle periodogram.

In the following analyses, we reduced computational load by fitting a single sine wave of fixed frequency $= 1/d$ (least-squares fit). For each fit, we extract the power as $P(f) = 1 - (\text{SSE}_{\text{sine}}/\text{SSE}_{\text{constant}})$ with $\text{SSE}_{\text{sine}}$ being the residuals (sum of squared errors) of the sine fit and $\text{SSE}_{\text{constant}}$ being the residuals (sum of squared errors) of a constant model assuming the mean of the data.

The power, hence, reflects how much of the velocity variation can be explained by the sinusoidal oscillation, or circadian rhythm.

**Targeted embedding using CCA**. We show that we can extract targeted embeddings from the spectral factors of the interaction networks that are better in predicting other properties of the individuals. To compute those targeted embeddings, we exactly follow the methodology outlined in Methods: 'Network age: from networks to spectral embeddings to CCA,' but for each property (days until death, time of peak activity, circadian rhythm, daytime and night-time velocities), we extract a one-dimensional embedding from the spectral factors that maximizes the correlation with this property using CCA.

**Prediction of other behavior-related measures**. We follow the same methodology as described in "Methods: 'Task prediction models and bootstrapping'" to evaluate how well the various age measures explain various properties of the individuals with a slight adjustment: We choose the identity function as the link function of the GLM and as the activation function of the neural network's final layer. We model the residual distribution as a normal distribution with constant variance.

We define the "days until death" as the number of days left in an individual's life on a given day (for a description of the automatically determined death dates see "Methods: 'Bayesian lifetime model'"). The time of peak activity and the rhythmicity of daily movement were calculated as described in "Methods: 'Definition of circadian rhythmicity.'"

For the daytime and night-time velocities we use the same data as for the circadian rhythmicity (see "Methods: 'Definition of circadian rhythmicity'"). For the daytime velocities, we use the mean of all collected velocities between

09:00–18:00 UTC of the 3-day rolling window; for night-time velocities, 21:00–06:00 UTC.

See Supplementary Table 1a for an overview of the scores of different models and targets.

**Future predictability**. We evaluate how well we can predict future task allocation using network age and biological age. To ensure that no information leak can occur, we only used supervised information from the past and test it on future data. To do this, we first calculate the spectral factors for the entire dataset as described in "Methods: 'Network age: from networks to spectral embeddings to CCA'." The factors are computed for each day separately, hence no information can leak from the past to the future. We then determine the mapping from spectral factors to network age using CCA, but only on a fixed range of days prior to the validation window. We fix the number of days in the train set to 12 days so that we always have approximately the same amount of training data independent of the number of days we predict into the future. Similar to the linear mapping given by CCA, we also determine the parameters of the regression model only on the train dataset from the same fixed time window. We train separate models for all viable ranges of dates and for prediction from one to 11 days into the future. The linear mapping given by the CCA and the predictive models is then applied to the spectral factors on the held-out validation set from a time after the training dataset (see Supplementary Fig. 21 for an overview about the data handling). For this analysis, we want to evaluate how well we can predict task allocation into the future. We estimate the effect size in $R^2_{\text{McF}}$ by calculating the 95% CI for the different time windows (see Supplementary Fig. 21, median improvement in $R^2_{\text{McF}} = 0.080$, 95% CI [0.055, 0.090], $N = 12$).

Because the two compared models are not nested, the likelihood ratio test does not apply here. We perform a paired binomial test using the null hypothesis that the improvement in mean-squared error in task allocation prediction is zero or less. We find that we can predict the task allocation of an individual 7 days into the future with a lower mean-squared error (paired binomial test, $p \ll 0.001$, $N = 55,390$) using network age.

This is mostly caused by older bees. For young bees, there is a low amount of variance in task allocation and network age is about as good in describing task allocation for young bees as biological age. Furthermore, we find that we can not reliably predict future task allocation for young bees, suggesting that either the social networks in this study are not predictive for this task or that the future task allocation of young bees is driven by other factors. See Supplementary Fig. 22 for an overview of the results.

The predictive power of network age cannot be fully explained by the bimodal distribution of network age and organization of stable "work groups" within the colony. We perform an additional analysis to reject this repeatability null hypothesis by comparing how well the task allocation prediction using networks works when simply shifting the prediction for the current day into the future. We find that a model fitted to predict future task allocation outperforms this null model considerably (see Supplementary Fig. 23).

**Ethics statement**. German law does not require approval of an ethics committee for studies involving insects.

**Reporting summary**. Further information on research design is available in the Nature Research Reporting Summary linked to this article.

## Data availability
The data (interaction networks and metadata) that support the findings of this study are available in zenodo with the identifier https://doi.org/10.5281/zenodo.4438013[93].

## Code availability
The code is available as open source software under the MIT license in zenodo with the identifier https://doi.org/10.5281/zenodo.441807[94].

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

## Acknowledgements

We are indebted to numerous students, in particular Fernando Wario Vázquez, Franziska Lojewski, Andreas Jörg, Leon Sixt, Hauke Mönck, Maria Sparenberg, Sascha Witte, Alexa Schlegel, Mathis Hocke, and Andreas Berg for providing hardware and software parts of the BeesBook system. We thank Peter Knoll and Randolf Menzel for providing the honey bee colony and valuable advice. Giovanni Galizia and Jake Graving gave critical feedback which helped improve the manuscript. Our work was supported by the HPC-Service of ZEDAT (Freie Universität Berlin) and the North-German Supercomputing Alliance who generously provided computing resources and storage capacity.

D.M.D. received funding from the Andrea von Braun Foundation, and the Elsa-Neumann-Scholarship. M.L.S. is a Simons Foundation Postdoctoral Fellow of the Life Sciences Research Foundation, and received funding from the Heidelberg Academy of Science and the Zukunftskolleg Mentorship Program. This project has received funding from the European Union's Horizon 2020 research and innovation program under grant agreement no. 824069. This work was, in part, funded by the Klaus Tschira Foundation Grant 00.300.2016. I.D.C. and M.L.S. acknowledge support from the Deutsche Forschungsgemeinschaft (DFG, German Research Foundation) under Germany's Excellence Strategy—EXC 2117—422037984 and I.D.C. acknowledges support from NSF Grant IOS-1355061 and Office of Naval Research Grants N00014-09-1-1074 and N00014-14-1-0635. K.S.T. acknowledges support of the Wissenschaftskolleg zu Berlin.

## Author contributions

Conceptualization: B.W., D.M.D., A.Z., T.L., M.L.S. and K.S.T.; methodology: B.W., D.M.D., A.Z. and T.L.; software: B.W., D.M.D.; resources, supervision: T.L., D.B. and I.D.C.; project administration: T.L.; data curation: B.W., D.M.D. and M.L.S.; writing: B.W., D.M.D., T.L., M.L.S., K.S.T. and I.D.C.; visualization: B.W. and D.M.D.

## Funding

## Competing interests

The authors declare no competing interests.
