## [Peer Review File · Nature Communications]

Reviewers' Comments:

Reviewer #1:

Remarks to the Author:

This is an innovative paper as well about the theory (talking about network age) as the methodology (identification of thousands of bees and analyses). Moreover, nice control experiments and analyses are present to confirm the original findings. The study present also interesting research perspectives about this new index. I am pleased to accept this paper for publication in Nature communication. I have just some minor remarks.

Generally, it would be nice to cite other studies than the ones belonging to Konstanz University or MPI. There are more pioneer or more fundamental works than for instance Aplin et al. 2015 or Strandburg-Peshkin et al. 2018

There are also other works on ants or bees networks than the ones of Pinter-Wollman (See Deneubourg, Theraulaz, Franks, etc.)

Lines 72–74: But some works already exist about identifying and following specific workers. Please cite them. It is also a broad question in humans with the elderly.

Figure 1: 'a single number' to refer to network age per bee is misleading here as it indicates to me one value per bee on the total of observations (lifetime). Add daily.

Line 126: The reasoning seems quite tautological here as network age is coming from interaction patterns in a place where tasks are realised and you compared task allocation to network age. Please explain why it is not so trivial/logical to find a correlation between the two variables. I know you tried to do it from lines 128 to 132 but please give more arguments (see for instance lines 141–142, 218–219) as this correlation is the core of your study.

Line 145 'marginal': please explain a bit more here. Add one sentence of explanation.

Line 232: please give references here about risk factors of disease transmission

Line 254 : see Münch D, Amdam GV, Wolschin F. 2008 Ageing in a eusocial insect: molecular and physiological characteristics of life span plasticity in the honey bee. *Funct. Ecol.* 22, 407–421.

(doi:10.1111/j.1365-2435.2008.01419.x) or Quque M, Benhaim-Delarbre M, Deneubourg J-L, Sueur C, Criscuolo F, Bertile F. 2019 Division of labour in the black garden ant (*Lasius niger*) leads to three distinct proteomes. *J. Insect Physiol.* , 103907. (doi:10.1016/j.jinsphys.2019.103907), for internal factors

Line 282: see these works:

- Baker N, Wolschin F, Amdam GV. 2012 Age-related learning deficits can be reversible in honeybees *Apis mellifera*. *Exp. Gerontol.* 47, 764–772. (doi:10.1016/j.exger.2012.05.011)

- Mersch DP, Crespi A, Keller L. 2013 Tracking individuals shows spatial fidelity is a key regulator of ant social organization. *Science* 340, 1090–1093.

- Amdam GV, Aase ALTO, Seehuus S-C, Kim Fondrk M, Norberg K, Hartfelder K. 2005 Social reversal of immunosenescence in honey bee workers. *Exp. Gerontol.* 40, 939–947. (doi:10.1016/j.exger.2005.08.004)

PS: I know there should be some space limitations, please not cite all references I give you but use the ones reinforcing your perspectives.

You used only one colony in your colony. Please explain how your results might be generalised to other colonies or why the studied colony is as other colonies, etc.

I am sorry, but I did not find the information about how you marked a bee. Maybe just give more information at the line 540.

Line 757: Do you mean S17B?

I think it would be nice to have a figure in the figure mixing fig SI2 and fig SI6B

fig SI12 : please indicate what the three colors (green-orange-blue) are, i.e. the three clusters.

Cédric Sueur

Reviewer #2:

Remarks to the Author:

This manuscript uses detailed, long-term individual tracking and interaction networks in honey bee colonies to generate a novel metric (network age) that they demonstrate to have strong predictive power for several aspects of worker behavior. Overall, the manuscript is technically impressive, well-written, and provides some novel insights. The paper highlights the strong correlation between task performance, space-use, and social interactions in honey bees in a detailed and compelling (if not entirely unexpected) way. The most novel and interesting findings are that patterns of social interactions are more predictive of task performance than biological age, and that this predictive power is, at least to some degree, independent of spatial dynamics.

While I think the manuscript is strong and will be of broad interest, there are some aspects of the manuscript that could be improved, detailed below.

Specific comments

L11: Suggest "repeatability" rather than "stability"

L35: Perhaps "more complete" instead of "entirety" here, since even perfect visual tracking will potentially miss non-visual interactions, etc

L113-114: Is this example real, or hypothetical? I.e., do trophallactic interactions have the strongest weight in the CCA?

L125-133: While technically possible it would be quite surprising if this metric of network age were not correlated with social interactions, given that it's intentionally structured to correlate with space-use. But the question about the degree of this correlation (rather than the existence/direction) seems richer and potentially more novel, especially in the context of looking at the relative predictive power of interaction network vs biological age (i.e., Fig 2B) below, so I suggest editing to highlight and further expand on this aspect.

L174-185: It seems like the # of clusters was fixed (at 3) if I'm understanding SI 3.2 right? How was this number of clusters chosen?

L186: Does averaging this transition across individual in cohorts make this transition look more gradual that it is at the individual scale? Possible to estimate the rate of transition in network age and compare to the cohort-averaged rates? Also obviously "gradual" is a relative matter, but a transition time on the order of a few days is, I think, more rapid than would be suggested by biological age-based cohort studies.

L223-225: Is it possible that behavior is still strongly tied to space, but that spatial information is just more fine-grained than picked up by regional annotations? E.g., perhaps whether a worker spend more time in the "core" of the brood area vs. on the "periphery" might be very predictive and would be potentially picked up by the network age metric (or alternatively, there are spatial regions associated with brood of different ages, etc), but these local spatial variations are treated the same by a binary regional annotation of task-associated spatial locations? Not sure if it's something you could directly address with the data here, but might be worth at least addressing.

L217: I read this sentence as claiming that network age has predictive power independent of spatial location. If that's the claim, I think this sentence should go after the next paragraph, which gives more direct evidence for it. Or rather, is the point here that network age doesn't just predict location, but also aspects of behavior (e.g, locomotion) within those locations?

L251-255: As noted above, I think some clarity on how especially the number of clusters was identified would be illuminating in comparing these findings to the Richardson et al. paper

cheers,
James Crall

Reviewer #3:

Remarks to the Author:

This creative and powerful method has a great deal of potential. We are very excited by the method the authors develop to condense multiple types of social interactions into a single, meaningful metric of behavior (in this study, a proxy for the tasks/roles of individual honey bees within a colony over time), but we are somewhat less excited by the biological insights that the method offers in the current manuscript. This is a really cool method, and we see many possible extensions to other social systems and we are confident that the author's approach will be adopted by other researchers possessing multiple types of social interaction data. This method, like any other network method, is only as good as the input fed into it (and we do have some concerns about the generation of trophallaxis networks, as we state below), but we see a huge upside future for the techniques used here. We recognize the incredible quantity and complexity of the data that the authors collected, and the presentation of their complicated methods for analyzing these data is clear and informative. The authors undertake a number of clever tests of the efficacy of their method (e.g. testing the ability of network age to predict other independently recorded behaviors, comparisons of the method with and without added spatial information, etc).

The particular dataset used as an example here presents some problems that should not take focus away, in the editorial evaluation of this paper, from the exciting development of new methods by these authors.

The specific application to honey bees raises questions about experimental design that impinge a little on the strong impression made by the network model. This is a very strong methods paper with a pilot bee study included as proof of concept. The contributions of the study to understanding honey bee division of labor is somewhat limited. The authors identify exciting avenues for future research (e.g. identifying the mechanisms that lead to bimodal development patterns amongst workers - a pattern that has been previously shown in bees, but was confirmed by their network age measure), but the authors stop short of testing explicit hypotheses. While collecting additional data (e.g. physiological, genetic data) would be beyond the scope of the study, we do feel as though the authors could dive more deeply into linking the complex social data that they collected to task allocation in honey bees. For instance, a major take-home of the manuscript is that network age is more informative than spatial information alone when describing task-associated behaviors. However, the authors do not explore how exactly social behaviors differ between individuals that differ by network age. In other words, what does the social context of bee interactions tell us about task allocation that previous approaches miss? This question is not addressed directly. We are certainly convinced that network age is capturing something interesting, but we would be very interested to see further exploration of what, biologically, network age is extracting. We are not experienced with spectral decomposition or canonical correlation analyses of networks, but would it be possible to identify features of the different input networks that disproportionately contribute to the final measure of network age? Even simple descriptive statistics as to how the different interaction networks and/or specific network measures contribute to network age would help to understand what network age is extracting about the biology/social behavior of individuals.

There are some concerns relating to the implied ethogram for the bee behavior. For the purposes of this paper, these concerns aren't lethal, but the authors should acknowledge in the paper that their

goal is to use the data to demonstrate how the method works more than testing detailed hypotheses. Several types of trophallactic interactions are conflated into a single measure. These include forager-nectar storer events, solicitation of food by nurses, and passage of nectar among foragers. Because these interactions occur in different locations within the nest, there is a possible confound with the spatial distribution measure. Trophallaxis is a directional behavior (donor and recipient) and also in some cases the donor is the initiator (returning foragers) while in other cases the recipient is the initiator (nurses seeking food) a directed network approach might give much more information and could be proposed for future research. Similarly, pollen and nectar foraging are different tasks and the variance in chronological age in the foragers may reflect this variance in role, which isn't assessed.

There is a very rich literature on division of labor in honeybees that has developed since Seeley 1982 and Johnson 2010. It has become apparent that tasks are much more finely divided among bees, with behavioral groupings like undertakers (necrophoric bees), guards, soldiers, fanners, pollen foragers, and nectar foragers becoming the focus of research. The relatively coarse grained view of division of labor taken in this paper misses an opportunity to use the associational and spatial data from this study to investigate variance in worker developmental pathways.

Also of concern from the point of view of a study on bees is that the sample size is $n = 1$ colony, making it impossible to assess intercolonial variance. We realize that this follows older approaches of often using only one colony in a study, but those studies were conducted decades ago and contemporary experimental design recognizes the value of replication at the colony level. We recognize that the focus on a single colony is in part determined by the intensive nature of the data collection, and we appreciate that the authors specifically identify the limitation of their method for directly comparing networks that do not share individuals (L269). It could be worth discussing how one might use the method in a larger, replicated experimental design.

Other comments:

L17. In its current wording, it is unclear whether you are referring to individual- or group-level 'functions' in this sentence.

L44. Change 'internal physiological factors' to 'internal factors', or keep physiological factors but provide separate references for genetic contributions. Genetics do not fall neatly into physiological factors, as currently written.

L68. The idea that networks of interactions are driven by more than spatial processes alone feels like a critical part of this interesting work. It would be very helpful to give examples/summarize (perhaps from these same citations that you list here) what these other, non-spatial influences might be.

L76. Would it be equally correct to frame this question as how an individual's role may contribute to its social network?

L78. Please be more specific about what you mean by "extract meaning".

Figure 1 is an excellent visual representation of your creative method.

L96. There is a discrepancy between these dates and the dates reported in SI 1.1 (2016-07-24 to 2016-09-19). Please clarify.

L98. How many of the 1921 individuals were present in each day of network data collection? A plot showing the number of bees over time and/or a histogram of the total number of days recorded per bee would help readers get a sense for the makeup of your networks.

L106. Your supplemental methods describe criteria for removing interactions from analysis

(occlusions, etc) so this does not seem like an accurate statement. We are not suggesting that the presumably small number of discarded/missed interactions would change the conclusions of the paper, but this sweeping statement does not seem to be true or necessary.

L113. You give the hypothetical example of CCA's weighting trophallactic interactions more when extracting information from the social networks, but you do not seem to report which features of the bee networks were most important in your actual application of the CCA. This would be very interesting to know.

L119. It is unclear from this sentence (and even when referring to the supplement) whether this focal period that you are referring to is a single day (it is clear that spectral decomposition of multimodal networks occurs on a daily basis) or over the entirety of your sampling period. Please clarify.

Why z-transform ages in Figure 2B?

L161. This is very cool that the method is robust to subsampling!

L214. It is very interesting that you test the efficacy of network age for describing behavioral differences by using independently recorded behaviors (nightly average speed, peak activity, etc), but it would be very helpful to the reader to know why these behaviors were chosen? Are there specific reasons/hypotheses why you would expect these measures to relate to a measure of task-allocation?

L223. This seems like a very nice test of the added information of network age over space use for describing task allocation, but see our above comments about wanting to know what it is about the social interactions of bees that improves our understanding of division of labor. This is very cool but we want to know more about what is being extracted.

Reviewer #1 (Remarks to the Author):

This is an innovative paper as well about the theory (talking about network age) as the methodology (identification of thousands of bees and analyses). Moreover, nice control experiments and analyses are present to confirm the original findings. The study present also interesting research perspectives about this new index. I am pleased to accept this paper for publication in Nature communication. I have just some minor remarks.

Thank you. We appreciate the positive feedback!

*Generally, it would be nice to cite other studies than the ones belonging to Konstanz University or MPI. There are more pioneer or more fundamental works than for instance Aplin et al. 2015 or Strandburg-Peshkin et al. 2018
There are also other works on ants or bees networks than the ones of Pinter-Wollman (See Deneubourg, Theraulaz, Franks, etc.)*

Thank you for these suggestions. In this paragraph we wanted to focus on works that use descriptors derived from the social network to elucidate the role of an individual. We agree that we did not state that clearly enough and changed the phrasing. We also noticed that the Sosna et al. paper did not fit into that view point and replaced it with a paper by Senova-Franks et al. Unfortunately, we're already over the limit of citations, and we hope that the fundamental works regarding social network analysis in general that you had in mind are covered by the review papers referenced in the preceding sentence.

Lines 72–74: But some works already exist about identifying and following specific workers. Please cite them. It is also a broad question in humans with the elderly.

We believe we've cited the relevant honey bee papers (Naug 2008, Crall et al. 2018, Blut et al. 2017, Gernat et al. 2018, Hasenjager et al. 2020, Bozek et al. 2020, etc.) as this paragraph is mainly concerned with honey bees. We have changed the sentence to reflect that (i.e. "individual honey bees" instead of "individuals") and also added the Siegel et al 2013 citation in the preceding paragraph as another example of a manual tracking study in honey bees. We do have a more general paragraph up above that we've extended. We're limited on the total number of citations, but please let us know if we've missed other important and related work that should definitely be cited.

While we would like to go into the importance of social networks for the elderly, we're already above the citation limit, and feel it is currently beyond the scope of this paper. Thus we've decided to focus on references on social insects in this section

Figure 1: 'a single number' to refer to network age per bee is misleading here as it indicates to me one value per bee on the total of observations (lifetime). Add daily.

We changed it to "single number per day".

Line 126: The reasoning seems quite tautological here as network age is coming from interaction patterns in a place where tasks are realised and you compared task allocation to network age. Please explain why it is not so trivial/logical to find a correlation between the two variables. I know you tried to do it from lines 128 to 132 but please give more arguments (see for instance lines 141–142, 218–219) as this correlation is the core of your study.

Yes, we expect a good measure of interaction patterns to correlate with the locations of the bees. But we need to show that network age IS a good measure and can extract the relevant information, so it's important to verify. The analysis also allows us to compare different ways of extracting network age (e.g. PCA, multimodal networks), how many dimensions are needed to capture this relationship (1D vs multidimensional representation), and compare to biological age.

We have added an introductory sentence to state our expectations more clearly. However, we do agree that this is only a necessary verification and not by itself sufficient. Therefore we later show that network age can also predict other behaviours that are not direct measures of space use.

Line 145 'marginal': please explain a bit more here. Add one sentence of explanation.

We mean marginal compared to the difference of the predictiveness of biological age and network age (a mean improvement of .34 in Pseudo R² vs .05 resp. .08 for the two and three dimensional variants). We revised the sentence accordingly.

Line 232: please give references here about risk factors of disease transmission

We now give proper references in the discussion.

*Line 254 : see Münch D, Amdam GV, Wolschin F. 2008 Ageing in a eusocial insect: molecular and physiological characteristics of life span plasticity in the honey bee. *Funct. Ecol.* 22, 407–421. (doi:10.1111/j.1365-2435.2008.01419.x) or Quque M, Benhaim-Delarbre M, Deneubourg J-L, Sueur C, Criscuolo F, Bertile F. 2019 Division of labour in the black garden ant (*Lasius niger*) leads to three distinct proteomes. *J. Insect Physiol.* , 103907. (doi:10.1016/j.jinsphys.2019.103907), for internal factors*

Line 282: see these works:

- Baker N, Wolschin F, Amdam GV. 2012 Age-related learning deficits can be reversible in honeybees *Apis mellifera*. *Exp. Gerontol.* 47, 764–772. (doi:10.1016/j.exger.2012.05.011)
- Mersch DP, Crespi A, Keller L. 2013 Tracking individuals shows spatial fidelity is a key regulator of ant social organization. *Science* 340, 1090–1093.
- Amdam GV, Aase ALTO, Seehuus S-C, Kim Fondrk M, Norberg K, Hartfelder K. 2005 Social reversal of immunosenescence in honey bee workers. *Exp. Gerontol.* 40, 939–947. (doi:10.1016/j.exger.2005.08.004)

PS: I know there should be some space limitations, please not cite all references I give you but use the ones reinforcing your perspectives.

Thank you for these references. We've extended the section in the introduction that covers the internal and external influences on behavior/aging and we've now included the Münch et al. 2008 paper. While we'd like to reference more works, we can only add more citations if absolutely necessary to conform with the submission guidelines.

You used only one colony in your colony. Please explain how your results might be generalised to other colonies or why the studied colony is as other colonies, etc.

We have elaborated the reasons why we think that our results are transferable to other colonies in the discussion and now include suggestions on how to perform hypothesis testing using network age in future studies.

I am sorry, but I did not find the information about how you marked a bee. Maybe just give more information at the line 540.

We added more details in the supplemental and made it explicit that we used the same marking procedure as in Wario et al., 2015.

Line 757: Do you mean S17B?

Thank you, we fixed the typo.

I think it would be nice to have a figure in the figure mixing fig SI2 and fig SI6B

After discussing this, we were unable to see what a figure mixing fig SI2 and fig SI6B would show. Could you please expand upon this? We have added a figure that gives more details about the number of individuals that we observed over the days of the focal period. Please have a look at the new figure in the Supplemental Information (SI 1.1) and see whether you'd need additional information.

fig SI12 : please indicate what the three colors (green-orange-blue) are, i.e. the three clusters.

We added an explanatory sentence.

Cédric Sueur

Reviewer #2 (Remarks to the Author):

This manuscript uses detailed, long-term individual tracking and interaction networks in honey bee colonies to generate a novel metric (network age) that they demonstrate to have strong predictive power for several aspects of worker behavior. Overall, the manuscript is technically impressive, well-written, and provides some novel insights. The paper highlights the strong correlation between task performance, space-use, and social interactions in honey bees in a detailed and compelling (if not entirely unexpected) way. The most novel and interesting findings are that patterns of social interactions are more predictive of task performance than biological age, and that this predictive power is, at least to some degree, independent of spatial dynamics.

While I think the manuscript is strong and will be of broad interest, there are some aspects of the manuscript that could be improved, detailed below.

Thank you for the positive comments.

Specific comments

L11: Suggest “repeatability” rather than “stability”

Thank you for the suggestion, but we feel stability is a more accurate description than repeatability for the following reasons. While we calculate the repeatability of network age, here we are offering an interpretation of our results: We believe that the high repeatability in network age indicates a high repeatability in task allocation. That, in combination with the fact that bees even in same-aged cohorts stuck to one of a handful of developmental trajectories (as opposed to dynamically switching back and forth), gives us a strong conviction that the task allocation on the individual level doesn't shift rapidly and is thus stable (in a more abstract sense of the word than would be indicated by a specific measure, such as repeatability.)

L35: Perhaps “more complete” instead of “entirety” here, since even perfect visual tracking will potentially miss non-visual interactions, etc

We agree. We removed the word “entirety” and added “more comprehensively”.

L113-114: Is this example real, or hypothetical? I.e., do trophallactic interactions have the strongest weight in the CCA?

We have added another analysis in SI 1.8 that looks at the contribution of the interaction networks. We noticed that the proximity contact network had the highest contribution (to network age 1D) and changed the sentence here accordingly.

L125-133: While technically possible it would be quite surprising if this metric of network age were not correlated with social interactions, given that it's intentionally structured to correlate with space-use. But the question about the degree of this correlation (rather than the existence/direction) seems richer and potentially more novel, especially in the context of looking at the relative predictive power of interaction network vs biological age (i.e., Fig 2B) below, so I suggest editing to highlight and further expand on this aspect.

We added an introduction to the paragraph where we explain that we do indeed expect network age to be correlated to our measure of space use (and why). This hopefully makes it a bit clearer that while we would indeed be surprised if that didn't hold true, it's still necessary to confirm and helpful when comparing the effect of different interaction types or variants of network age (PCA vs. CCA).

L174-185: It seems like the # of clusters was fixed (at 3) if I'm understanding SI 3.2 right? How was this number of clusters chosen?

We fixed the number at 3 as that is the smallest number that still separates the second transition for all tested cohorts for visualization purposes. As the exact timing and shape of the developmental paths vary even across our cohorts and might be specific to our setup, we do not think that this specific choice of three clusters is something that will necessarily transfer to other setups. However, we believe the ability to use network age to distinguish different developmental pathways will be transferable and should be adjusted based on the life histories of the specific social organism.

We added an explanatory sentence in the main text and SI to clarify our choice. We've also added an additional plot with N=5 clusters in the SI. With additional clusters, the trajectories for the old upper blue cluster and the old lower green cluster are both split into two further sub clusters of approximately equal size.

L186: Does averaging this transition across individual in cohorts make this transition look more gradual that it is at the individual scale? Possible to estimate the rate of transition in network age and compare to the cohort-averaged rates? Also obviously "gradual" is a relative matter, but a transition time on the order of a few days is, I think, more rapid than would be suggested by biological age-based cohort studies.

While we do average the location for the heatmap plots, we do not average the network age development for the cohort trajectory where the individual points change from low network age to high network age. Also note that with "gradual", we don't mean slow, but rather that the transition is not instantaneous. Despite this, measuring the rate of transition is certainly something that would be great to explore in future works, in particular with more distinct cohorts and colonies.

L223-225: Is it possible that behavior is still strongly tied to space, but that spatial information is just more fine-grained than picked up by regional annotations? E.g., perhaps whether a worker spend more time in the "core" of the brood area vs. on the "periphery" might be very predictive and would be potentially picked up by the network age metric (or alternatively, there are spatial regions associated with brood of different ages, etc), but these local spatial variations are treated the same by a binary regional annotation of task-associated spatial locations? Not sure if it's something you could directly address with the data here, but might be worth at least addressing.

Yes, we believe that behavior is strongly tied to space, and maybe even more so than our analysis based on the rather coarse location annotations suggests. But we also think that it's very unlikely that there's nothing more than just location that determines behavior. Note that instead of categorical labels (such as "nurse" or "MAB") we use the distribution of time spent in our annotated regions (e.g. "50% brood, 50% honey storage"), which alleviates this problem. We changed the "highlights" to "suggests" in the conclusion of this paragraph to reflect this.

L217: I read this sentence as claiming that network age has predictive power independent of spatial location. If that's the claim, I think this sentence should go after the next paragraph,

which gives more direct evidence for it. Or rather, is the point here that network age doesn't just predict location, but also aspects of behavior (e.g, locomotion) within those locations?

The latter: We wanted to test whether network age also predicts behaviors that do not directly measure a bee's location unlike our spatial task allocation proxy. We rephrased the paragraph to make our intention clearer.

L251-255: As noted above, I think some clarity on how especially the number of clusters was identified would be illuminating in comparing these findings to the Richardson et al. paper

The Richardson et al paper identified three non overlapping communities in their social maturity index. This index is comparable to network age, but please note that we did not directly cluster network age but instead clustered the development of network age over the life of individuals, so the number of clusters in our case (clusters of life trajectories) does not directly compare to the number of clusters in the Richardson et al. paper (clusters of maturity). We also see a bimodality in network age similar to the one in the social maturity index of Richardson et al. and can also roughly identify three groups of network age values (low, mid-high, and high; see the three panels of figure 2A), but we think it's more valuable to think about network age as a continuous value and did not attempt to directly assign individuals into distinct network age groups. We revised this section to make this point clearer and also explain how we chose the number of clusters in SI 3.2).

*cheers,
James Crall*

Reviewer #3 (Remarks to the Author):

This creative and powerful method has a great deal of potential. We are very excited by the method the authors develop to condense multiple types of social interactions into a single, meaningful metric of behavior (in this study, a proxy for the tasks/roles of individual honey bees within a colony over time), but we are somewhat less excited by the biological insights that the method offers in the current manuscript. This is a really cool method, and we see many possible extensions to other social systems and we are confident that the author's approach will be adopted by other researchers possessing multiple types of social interaction data. This method, like any other network method, is only as good as the input fed into it (and we do have some concerns about the generation of trophallaxis networks, as we state below), but we see a huge upside future for the techniques used here. We recognize the incredible quantity and complexity of the data that the authors collected, and the presentation of their complicated methods for analyzing these data is clear and informative. The authors undertake a number of clever tests of the efficacy of their method (e.g. testing the ability of network age to predict other independently recorded behaviors, comparisons of the method with and without added spatial information, etc).

The particular dataset used as an example here presents some problems that should not take focus away, in the editorial evaluation of this paper, from the exciting development of new methods by these authors.

The specific application to honey bees raises questions about experimental design that impinge a little on the strong impression made by the network model. This is a very strong methods paper with a pilot bee study included as proof of concept. The contributions of the study to understanding honey bee division of labor is somewhat limited. The authors identify exciting avenues for future research (e.g. identifying the mechanisms that lead to bimodal development patterns amongst workers - a pattern that has been previously shown in bees, but was confirmed by their network age measure), but the authors stop short of testing explicit hypotheses. While collecting additional data (e.g. physiological, genetic data) would be beyond the scope of the study, we do feel as though the authors could dive more deeply into linking the complex social data that they collected to task allocation in honey bees. For instance, a major take-home of the manuscript is that network age is more informative than spatial information alone when describing task-associated behaviors. However, the authors do not explore how exactly social behaviors differ between individuals that differ by network age. In other words, what does the social context of bee interactions tell us about task allocation that previous approaches miss? This question is not addressed directly. We are certainly convinced that network age is capturing something interesting, but we would be very interested to see further exploration of what, biologically, network age is extracting. We are not experienced with spectral decomposition or canonical correlation analyses of networks, but would it be possible to identify features of the different input networks that disproportionately contribute to the final measure of network age? Even simple descriptive statistics as to how the different interaction networks and/or specific network measures contribute to network age would help to understand what network age is extracting about the biology/social behavior of individuals.

Thank you for the positive comments.

We had refrained from including an analysis of which features from the social networks were used by the CCA before. As you rightly state, our interaction matrices are coarse (e.g. combining different types of trophallactic into one network) and this might have an effect on

the features used by the CCA so we don't want to make absolute statements such as "proximity is more important than trophallaxis" with our data that could potentially not apply to other setups.

However, we do agree that showing how such an analysis could be performed with our method as well as showing how our CCA assigns weights to our specific input matrices is a valuable addition to the paper and added such an analysis in SI 1.8.

There are some concerns relating to the implied ethogram for the bee behavior. For the purposes of this paper, these concerns aren't lethal, but the authors should acknowledge in the paper that their goal is to use the data to demonstrate how the method works more than testing detailed hypotheses.

Yes, we agree that the method is an integral part of our work, but we also believe that our biological findings generalise to other colonies, which we have now elaborated on in the discussion. In addition, we also added an explanation of how future detailed hypothesis testing could be performed.

Several types of trophallactic interactions are conflated into a single measure. These include forager-nectar storer events, solicitation of food by nurses, and passage of nectar among foragers. Because these interactions occur in different locations within the nest, there is a possible confound with the spatial distribution measure. Trophallaxis is a directional behavior (donor and recipient) and also in some cases the donor is the initiator (returning foragers) while in other cases the recipient is the initiator (nurses seeking food) a directed network approach might give much more information and could be proposed for future research.

You are right that we neither distinguish between different types of trophallaxis, nor do we label the donor and receiver. Both of these distinctions could improve the network age measure. In general, we believe that a more fine-grained resolution of different interaction behaviors could be a tremendous gain for our resulting descriptor - we now highlight that point in the discussion. We also expanded details in the supplement to clarify that we do not distinguish between different kinds of trophallactic interactions. We demonstrate that our method works for directed networks using the speed-transfer-networks as an example where we separated the directed graphs into two inputs used for the spectral decomposition.

Similarly, pollen and nectar foraging are different tasks and the variance in chronological age in the foragers may reflect this variance in role, which isn't assessed.

In our feeder experiment, we only supplied a sugar solution and still saw foragers with a very high variance in biological age (see fig. 2B), so we think that even if the difference of nectar and pollen foraging tasks may be a contributing factor in the age variance of the foragers, it will not be enough to explain the total effect. Much prior research shows that while pollen foragers typically forage earlier in life than nectar foragers, the age of first foraging still diverges widely within each forager subset. In the results we've now clarified that the bees observed at the feeding site were known nectar foragers.

There is a very rich literature on division of labor in honeybees that has developed since Seeley 1982 and Johnson 2010. It has become apparent that tasks are much more finely divided among bees, with behavioral groupings like undertakers (necrophoric bees), guards, soldiers, fanners, pollen foragers, and nectar foragers becoming the focus of research. The relatively coarse grained view of division of labor taken in this paper misses an opportunity to use the associational and spatial data from this study to investigate variance in worker developmental pathways.

We agree that the data we collected offers much more than what we could explore in this work. For example, when looking at two random bees with very similar network age but different spatial distributions, we noticed that both bees would spend their whole day engaged in washboarding behavior and cleaning the glass of our observation hive. Looking into how well network age can resolve more fine grained behaviors would be a great venue for future work.

Also of concern from the point of view of a study on bees is that the sample size is $n = 1$ colony, making it impossible to assess intercolonial variance. We realize that this follows older approaches of often using only one colony in a study, but those studies were conducted decades ago and contemporary experimental design recognizes the value of replication at the colony level. We recognize that the focus on a single colony is in part determined by the intensive nature of the data collection, and we appreciate that the authors specifically identify the limitation of their method for directly comparing networks that do not share individuals (L269). It could be worth discussing how one might use the method in a larger, replicated experimental design.

You are right and in our conclusion we now discuss why we believe our findings are valid even though we only looked at one colony and how we believe hypothesis testing in replicated experimental designs should be performed.

Other comments:

L17. In its current wording, it is unclear whether you are referring to individual- or group-level 'functions' in this sentence.

We rephrased the sentence to make clear that we mean the individual's function.

L44. Change 'internal physiological factors' to 'internal factors', or keep physiological factors but provide separate references for genetic contributions. Genetics do not fall neatly into physiological factors, as currently written.

We changed it to "internal factors".

L68. The idea that networks of interactions are driven by more than spatial processes alone feels like a critical part of this interesting work. It would be very helpful to give examples/summarize (perhaps from these same citations that you list here) what these other, non-spatial influences might be.

Thank you, we now quickly summarize the influences from the cited works in this sentence.

L76. Would it be equally correct to frame this question as how an individual's role may contribute to its social network?

Yes. We rephrased the sentence to remove the directionality and changed it to the "relationship between an individual's social network and its lifetime role".

L78. Please be more specific about what you mean by "extract meaning".

We rephrased the paragraph to make our intention clearer.

Figure 1 is an excellent visual representation of your creative method.

Thank you!

L96. There is a discrepancy between these dates and the dates reported in SI 1.1 (2016-07-24 to 2016-09-19). Please clarify.

The wording was indeed confusing - the marking period began before the focal period. We elaborated on the used dates in the SI and added a reference in the main text.

L98. How many of the 1921 individuals were present in each day of network data collection? A plot showing the number of bees over time and/or a histogram of the total number of days recorded per bee would help readers get a sense for the makeup of your networks.

We added a supplemental figure that shows the number of bees per day in our focal period as well as a histogram of the number of days per bee.

L106. Your supplemental methods describe criteria for removing interactions from analysis (occlusions, etc) so this does not seem like an accurate statement. We are not suggesting that the presumably small number of discarded/missed interactions would change the conclusions of the paper, but this sweeping statement does not seem to be true or necessary.

We removed the two “all” quantifiers.

L113. You give the hypothetical example of CCA’s weighting trophallactic interactions more when extracting information from the social networks, but you do not seem to report which features of the bee networks were most important in your actual application of the CCA. This would be very interesting to know.

We added another analysis in the supplemental and show which weights the CCA assigns to the different social networks (SI 1.8).

L119. It is unclear from this sentence (and even when referring to the supplement) whether this focal period that you are referring to is a single day (it is clear that spectral decomposition of multimodal networks occurs on a daily basis) or over the entirety of your sampling period. Please clarify.

We changed it to “over the 25 days of the focal period” to make it clearer that we mean the whole interval.

Why z-transform ages in Figure 2B?

Network age and biological age are not on the same scale. Comparing their raw variances would therefore be misleading. So we z-transform them in order to be able to compare the mean and variance of the numbers. We added a sentence on why we apply the z-transformation.

L161. This is very cool that the method is robust to subsampling!

L214. It is very interesting that you test the efficacy of network age for describing behavioral differences by using independently recorded behaviors (nightly average speed, peak activity, etc), but it would be very helpful to the reader to know why these behaviors were chosen? Are there specific reasons/hypotheses why you would expect these measures to relate to a measure of task-allocation?

We rephrased the paragraph and added a short explanation for why we chose these features: We believe they are related to task allocation as foraging is an inherently diurnal activity with foraging only possible at day time.

L223. This seems like a very nice test of the added information of network age over space use for describing task allocation, but see our above comments about wanting to know what it is about the social interactions of bees that improves our understanding of division of labor. This is very cool but we want to know more about what is being extracted.

We have added such an analysis to the supplemental (SI 1.8, see our comment above).

Reviewers' Comments:

Reviewer #1:

Remarks to the Author:

Authors answered mostly to all my comments and the paper is now ready to be published in Nature Communications. Details authors added to the paper reinforced its quality. Such innovative research has its place in Nature Communications. I just come back on one comment I made to not only cite papers from their institute, particularly the one on culture spread and the one on leadership in mobile groups. Please add citations of other important contributors at such places as these contributors are attentive to such behaviour. Nice work!

Cédric Sueur

Reviewer #2:

Remarks to the Author:

The authors have sufficiently addressed all my concerns in this revised manuscript, and remain enthusiastic about this comprehensive and novel paper that I believe will be of broad interest.

cheers,

James Crall

Reviewer #3:

Remarks to the Author:

The authors have more than adequately addressed our concerns, and we feel that this valuable contribution needs no further revision.

Reviewer #1 (Remarks to the Author):

Authors answered mostly to all my comments and the paper is now ready to be published in Nature Communications. Details authors added to the paper reinforced its quality. Such innovative research has its place in Nature Communications. I just come back on one comment I made to not only cite papers from their institute, particularly the one on culture spread and the one on leadership in mobile groups. Please add citations of other important contributors at such places as these contributors are attentive to such behaviour.

Nice work!
Cédric Sueur

We agree, and have added more references on culture spread and leadership in mobile groups.

Reviewer #2 (Remarks to the Author):

The authors have sufficiently addressed all my concerns in this revised manuscript, and remain enthusiastic about this comprehensive and novel paper that I believe will be of broad interest.

cheers,
James Crall

Reviewer #3 (Remarks to the Author):

The authors have more than adequately addressed our concerns, and we feel that this valuable contribution needs no further revision.